# Vaccine-Acquired SARS-CoV-2 Immunity versus Infection-Acquired Immunity: A Comparison of Three COVID-19 Vaccines

**DOI:** 10.3390/vaccines10122152

**Published:** 2022-12-15

**Authors:** Marie I. Samanovic, Aaron L. Oom, Amber R. Cornelius, Sophie L. Gray-Gaillard, Trishala Karmacharya, Michael Tuen, Jimmy P. Wilson, Meron F. Tasissa, Shelby Goins, Ramin Sedaghat Herati, Mark J. Mulligan

**Affiliations:** NYU Langone Vaccine Center, Department of Medicine, New York University Grossman School of Medicine, New York, NY 10016, USA

**Keywords:** COVID-19, vaccines, SARS-CoV-2, immunity, delta, omicron

## Abstract

Around the world, rollout of COVID-19 vaccines has been used as a strategy to end COVID-19-related restrictions and the pandemic. Several COVID-19 vaccine platforms have successfully protected against severe SARS-CoV-2 infection and subsequent deaths. Here, we compared humoral and cellular immunity in response to either infection or vaccination. We examined SARS-CoV-2 spike-specific immune responses from Pfizer/BioNTech BNT162b2, Moderna mRNA-1273, Janssen Ad26.COV2.S, and SARS-CoV-2 infection approximately 4 months post-exposure or vaccination. We found that these three vaccines all generate relatively similar immune responses and elicit a stronger response than natural infection. However, antibody responses to recent viral variants are diminished across all groups. The similarity of immune responses from the three vaccines studied here is an important finding in maximizing global protection as vaccination campaigns continue.

## 1. Introduction

Three years after the identification of the novel coronavirus, SARS-CoV-2, in Wuhan, China, close to 640 million confirmed cases of coronavirus disease 2019 (COVID-19) have been reported globally, including over six and half million deaths [1,2], at the time of writing (November 2022). To respond to the pandemic, many nations have relied on restrictions in parallel with the deployment of vaccines. Out of the 11 vaccines currently recommended for use by the World Health Organization (WHO) [3,4], four are authorized or licensed for use in the United States [5,6,7,8,9]: the mRNA vaccine BNT162b2 (Pfizer-BioNTech COVID-19 vaccine), the mRNA vaccine mRNA-1273 (Moderna COVID-19 vaccine), the adenoviral-vectored vaccine Ad26.COV2.S (Janssen COVID-19 vaccine, also referred to as the Johnson & Johnson vaccine (J&J)), and more recently, the recombinant protein-based NVX-CoV2373 (Novavax COVID-19 vaccine).

All four vaccines licensed or authorized in the US provide robust protection against COVID-19-related severe illness and death. The rollout in the United States of the Pfizer-BioNTech, Moderna, and J&J at the end of 2020 or early in 2021 have made a substantial impact in controlling the pandemic [10,11,12]. SARS-CoV-2 has four key structural proteins: the Nucleocapsid Protein (N Protein), the Spike Protein (S Protein), the Matrix Protein (M Protein), and the Envelope Protein (E Protein). The S protein plays a key role in the receptor recognition and cell membrane fusion process and is the immunogenic protein for all four of the US approved COVID-19 vaccines. The S protein is composed of two subunits: S1 (containing a receptor-binding domain) and S2 (mediating viral cell membrane fusion) [13,14].

Earlier in the pandemic, it was believed that fully vaccinated individuals (defined as 2 weeks after one dose of a J&J single-dose vaccine, or 2 weeks after a second dose of an mRNA two-dose vaccine series) and those previously infected with SARS-CoV-2 had low risk of symptomatic infection for at least six months. Before the emergence of omicron variants, infection with an earlier coronavirus strain was roughly 90 percent effective in preventing a symptomatic infection [15]. However, the emergence of variants of concern that can partially escape vaccine-induced immunity paired with naturally waning humoral immunity [16] has eroded the magnitude and the longevity of such protection.

In response to the erosion of vaccine protection, some countries have authorized third and even fourth vaccine doses as boosters, and the US has begun a bivalent prototype+BA.4-5 booster campaign that provides a fifth booster vaccination for some. However, throughout much of the developing world many remain unvaccinated due to a global rollout that remains slow and limited [17]. While initial doses of vaccine continue to be distributed throughout these regions, it is important to understand if particular vaccines should be favored by using the abundance of data generated within highly vaccinated countries.

Here we assess the humoral and cellular immune responses generated by the first three vaccines authorized and deployed in the US (Pfizer-BioNTech, Moderna, and J&J) in vaccinated individuals who had not been previously infected and compare them to the immune responses elicited by natural infection in unvaccinated individuals (convalescent patients). Overall, we show relatively few differences between the three vaccinated groups, with noteworthy exceptions including the higher binding antibody titers in mRNA1273 vaccinees, and the immunodominance of the S protein subunit 1 (S1) in vaccinated individuals but not convalescent participants. All three vaccines elicit a stronger response than natural infection 4 months post-vaccination or exposure to the virus.

## 2. Materials and Methods

### 2.1. Study Design

We conducted an observational study of adults who were fully vaccinated after receiving one of the COVID-19 vaccines available in the US in 2021, according to CDC guidelines (BNT162b2, mRNA-1273, or Ad26.COV2.S), or had a documented history of SARS-CoV-2 infection. Individuals with severe anemia or inability to comply with study procedures were excluded. One hundred and sixteen adults provided written consent for enrollment with approval from the NYU Institutional Review Board (protocols 18-02035 and 18-02037). Participants had blood drawn around 4 months post-vaccination (post-second dose for mRNA vaccines, post-first dose for Ad26.COV2.S vaccine) or post-COVID-19 (post-onset of symptoms). The median numbers of days ± standard deviations were: 124 ± 12 for BNT162b2; 126 ± 15 for mRNA-1273, 118 ± 20 for Ad26.COV2.S, and 120 ± 20 for convalescent participants.

Participant characteristics are summarized in Table 1, Appendix A. Participants self-reported medical history and this was verified by review of provider notes in the electronic medical record. All vaccinated individuals included in this study did not have a known history of SARS-CoV-2 infection or breakthrough infection. Their naive status was further confirmed by negative N-specific IgG ELISA (Figure 1A).

### 2.2. Blood Samples, Specimen Processing, and Storage

Venous blood was collected by standard phlebotomy. PBMCs were isolated from CPT vacutainers (BD Biosciences, Franklin Lakes, NJ, USA) and processed within four hours of collection. PBMCs were viably cryopreserved and thawed later for assays. Sera were collected in SST tubes (BD Biosciences, Franklin Lakes, NJ, USA) and frozen immediately at −80 °C.

### 2.3. ELISAs

Direct ELISA was used to quantify the neutralization titers for participants’ sera. Ninety-six well plates were coated with 1 μg/mL S1 protein (100 μL/well) or 0.1 μg/mL N protein diluted in phosphate-buffered saline (PBS) and were then incubated overnight at 4 °C (Sino Biological Inc., Chesterbrook, PA, USA, 40591-V08H and 40588-V08B). Plates were washed four times with 250 μL of PBS containing 0.05% Tween 20 (Thermo Fisher Scientific, Waltham, MA, USA) (PBS-T), and blocked with 200 μL PBS-T containing 5% non-fat milk at room temperature for 1 h. Sera were heat-inactivated at 56 °C for 1 h prior to use. Samples were analyzed in duplicate. Duplicates were diluted to a starting concentration of 1:50 (S1) or 1:100 (N) then serially diluted 1:3 in blocking solution. The final volume in all wells after dilution was 100 μL. After a 2 h incubation at room temperature, plates were washed four times with PBS-T. Horseradish peroxidase (HRP)-conjugated goat-anti human IgG (Southern BioTech, Birmingham, AL, USA, 2040-05) was diluted in blocking buffer (1:2000) and 100 μL was added to each well. Plates were incubated for 1 h at room temperature then washed four times with PBS-T. After developing for 5 min with TMB Peroxidase Substrate 3,3′,5,5′-Tetramethylbenzidine (Thermo Fisher Scientific, Waltham, MA, USA, 34029), the reaction was stopped with 1N hydrochloric acid (Fisher Scientific, Waltham, MA, USA, AC124210025). The optical density was determined by measuring the absorbance at 450 nm on a Synergy 4 (BioTek, Venusky, VT, USA) plate reader. To summarize data collected on individuals, the area under the response curve was calculated for each sample and endpoint titers were normalized using replicates of pooled positive control sera on each plate to reduce variability between plates.

### 2.4. IgG Avidity Assays

Ninety-six well plates were coated with 0.1 µg/mL S1 protein (100 µL/well) diluted in PBS overnight at 4 °C. Plates were washed four times with 250 µL PBS-T then blocked with 200 µL PBS-T containing 4% non-fat milk and 5% whey at RT for 1 h. Sera were heat-inactivated at 56 °C for 1 h prior to use. Samples were diluted to a starting concentration of 1:50 and added to the plates in quadruplicate, then serially diluted 1:3 in blocking solution. The final volume in all wells after dilution was 100 µL. After a 2 h incubation period at RT, plates were washed four times with PBS-T. PBS was then added to two dilution replicate sets and 6 M Urea to the other two dilution replicate sets. Plates were incubated for 10 min at RT before washing four times with PBS-T. Antibodies were detected and plates were developed and read as described above for ELISA.

Avidity was calculated by dividing the dilutions that gave an optical density value of 0.5 (Urea treatment/no Urea) × 100. Scores with theoretical values between 0 and 100% were generated.

### 2.5. Memory B Cell ELISpot Assays

ELISpot assays were used to detect antigen-specific IgG memory B cells using cryopreserved PBMCs. PBMCs were rapidly thawed in a 37 °C water bath and slowly added to 10 mL of warmed RPMI-1640 containing 10% fetal calf serum (FCS), 100 units/mL of penicillin G, and 100 μg/mL of streptomycin (Gibco, New York, NY, USA), referred to as R10, with 1:5000 dilution of recombinant DNase I (Roche, Basel, Switzerland, 04716728001). The cells were then counted using Countess and resuspended as 1.0 × 10^6^ cells/mL in stimulation media (R10 with 1:5000 dilution of DNase I and 1:2000 dilution of B-Poly-S (CTL-BPOLYS-200, ImmunoSpot) and 1 μL of 1 mM 2-Mercaptoethanol (Sigma, M3148) per mL of R10). A total of 1.0 × 10^6^ cells in 1 mL were then added to each well of a 24-well plate (Corning, Somerville, MA, USA, 3527) and incubated at 37 °C with 5% CO_2_ for 5 days. At least 18 h prior to day 6, sterile Millipore multiscreen-HA 96-well plates with an MCE membrane (Millipore, Burlington, MA, USA, MSHAN4B50) were coated with 2 μg/mL of target recombinant antigens—S1, S2, RBD, or N (Sino Biological Inc., Chesterbrook, PA, USA, 40591-V08H, 40590-V08B, 40592-V08H, and 40588-V08B), 10 μg/mL of donkey anti-human IgG capture antibody (Jackson ImmunoResearch Laboratory Inc., West Grove, PA, USA, 709-005-149), or 2.5 μg/mL of negative control Imject Mariculture keyhole limpet hemocyanin (KLH) (Thermo Fisher Scientific, Waltham, MA, USA, 77600). Coated plates were then incubated at 4 °C. On day 6, coated plates were washed four times with 200 μL of PBS-T and blocked for 2 h at 37 °C with 200 μL R10. Stimulated cultures from the 24-well plate were transferred into 5 mL sterile conical tubes and resuspended in R10 at 1.0 × 10^6^ cells/100 μL. A total of 50 μL of cells in R10 at 1.0 × 10^6^ cells/100 μL were added to the top row of specific antigen detection wells containing 150 μL R10 and 5 μL of cells to the first dilution wells for total IgG detection wells containing 195 μL R10. A 3-fold serial dilution was then performed three times. Total IgG detection wells were diluted 10-fold higher than the target antigen wells to facilitate spot-counting. Plates were incubated for 6 h at 37 °C with 5% CO_2_. Plates were then washed three times with 200 μL PBS followed by four times with 200 μL PBS-T. Biotinylated goat anti-human IgG (Jackson ImmunoResearch Laboratory Inc., West Grove, PA, USA, 709-065-098) was diluted 1:1000 in PBS-T with 2% FCS (Ab diluent) and 100 μL was added to the wells for overnight incubation at 4 °C. Plates were washed four times with 200 μL of PBS-T and incubated with 100 μL of Avidin-D-HRP conjugate (Vector Laboratories, Newark, CA, USA, A-2004) diluted 1:1000 in PBS-T with 2% FCS for 1 h at RT. Plates were washed four times with 200 μL PBS-T and three times with PBS. 100 μL of AEC substrate (3 amino-9 ethyl-carbazole; Sigma Aldrich, St. Louis, MO, USA, A-5754) was added. Plates were incubated at room temperature for 5 min and rinsed with water to stop the reaction. Developed plates were scanned and analyzed using an ImmunoSpot automated ELISpot counter (Cellular Technology Limited, Shaker Heights, OH, USA) or Cytation 7 (BioTek, Venusky, VT, USA).

### 2.6. Immunofluorescence-Based SARS-CoV-2 Live Virus Microneutralization Assays

Neutralization assays were performed in the ABSL3 facility of NYU Grossman School of Medicine (New York, NY, USA), in accordance with its Biosafety Manual and Standard Operating Procedures. Viral neutralization activities of plasma were measured in an immunofluorescence-based microneutralization assay by detecting the neutralization of infectious virus in cultured Vero E6 cells (African Green Monkey Kidney; ATCC, CRL-1586) or Vero E6-TMPRSS2-T2A-ACE2 cells (BEI, NR-54970). Cells were maintained according to standard ATCC protocols [18]. SARS-CoV-2 isolate USA-WA1/2020, deposited by the Centers for Disease Control and Prevention, was obtained through BEI Resources, NIAID, NIH (NR-52281, GenBank accession no. MT233526). SARS-CoV-2 delta variant (hCoV19/USA/PHC658/2021) and SARS-CoV-2 omicron BA.1 (hCoV-19/USA/GA-EHC-2811C/2021) were kindly provided by the Suthar Lab at Emory University. Viruses were passaged once in Vero E6 cells supplemented with 1 µg/mL of TPCK-Trypsin at a multiplicity of infection of 0.01 and harvested at 50% cytopathic effect. After harvest, virus was purified using a 25% sucrose cushion at 107,100× *g* (25,000 RPM with SW 32 Ti rotor) for 3–4 h and resuspended using PBS prior to infection.

Serial dilutions of heat-inactivated plasma, in duplicate, (56 °C for 1 h) were incubated with USA-WA1/2020 stock, B.1.167.2 delta stock or B.1.1.529 omicron stock (at fixed 1 × 10^6^ PFU/mL, 9.9 × 10^7^ PFU/mL, and 3.5 × 10^7^ PFU/mL, respectively) in DMEM (Corning, 10-013CV) supplemented with 1% of MEM Nonessential Amino Acid (NEAA) Solution, and 10 mM Hepes (ThermoFisher, Waltham, MA, USA, 15-630-080) for 1 h at 37 °C. One hundred microliters of the plasma-virus mix was then added to the cells and incubated at 37 °C with 5% CO_2_. Twenty-four hours post-infection, cells were fixed with 10% formalin solution (4% active formaldehyde) for 1 h, stained with 0.5 µg/mL anti-SARS-CoV-2 nucleocapsid antibody (ProSci, Phoenix, AZ, USA, 10-605), and 1 µg/mL goat anti-mouse IgG AF647 secondary antibody (ThermoFisher, Waltham, MA, USA, A32728) along with DAPI (1.24 µg/mL), and visualized by microscopy with the CellInsight CX7 High-Content Screening (HCS) Platform (ThermoFisher, Waltham, MA, USA) and high-content software (HCS), or the BioTek Cytation 7 Cell Imaging Multi-Mode Reader and Gen5 Image Prime software.

### 2.7. Activation-Induced Marker (AIM) and Intracellular Cytokine Staining (ICS) Analyses

Cryopreserved PBMCs were thawed and rested overnight at 37 °C in RPMI 1640 with L-glutamine (Corning, 10-041-CV) containing 10% FBS (ThermoFisher, Waltham, MA, USA, 10082-147), 2 mM L-glutamine (Fisher, Waltham, Massachusetts, USA, 25030-081), and 100 U/mL of penicillin–streptomycin (ThermoFisher, Waltham, MA, USA, 15-140-122). The following day, cells were stimulated with 0.6 nmol of each of the S1, S, and S+ PepTivator pools (Miltenyi, 130-127-048, 130-126-700, and 130-127-312, respectively) for 20 h at 37 °C with 1.5 × 10^6^ cells per well in a 96-well flat bottom plate. For the unstimulated control well, sterile water was used in place of the peptide pools. Monensin (ThermoFisher, Waltham, MA, USA, 00-4505-51) was added for the last 6 h of stimulation at a final concentration of 10 μM. After stimulation, cells were washed with PBS containing 10 mM EDTA at 37 °C for 5 min, followed by Fc-blockade and were stained as previously described [18]. Antibodies, clones, and catalog numbers are described in Appendix A. Analyses were performed using FlowJo.

## 3. Results

### 3.1. Study Participants

A total of 116 participants were enrolled for this study. To compare the antibody response between vaccine platforms, 95 individuals were enrolled between May and August 2021 following vaccination with Pfizer-BioNTech (*n* = 47), Moderna (*n* = 28), or J&J vaccines (*n* = 20) (Table 1 and Appendix A). Blood was collected from participants approximately 4 months after completion of the primary vaccine series (after two doses of mRNA vaccines or after one dose of Ad26.COV2.S vaccine, Appendix A). Participants did not have a documented history of COVID-19, as confirmed by direct Enzyme-Linked Immunosorbent Assay (ELISA) measuring anti-Nucleocapsid (anti-N) IgG antibodies in participant serum (Figure 1A). Vaccinated participants had not received booster vaccinations after completion of their primary vaccine series.

The immune response to vaccination was also compared to the responses from natural infection by enrolling individuals who were infected with SARS-CoV-2 between March and April 2020 (Convalescent, *n* = 21). Infection was confirmed by a nucleic acid amplification test during the acute phase of infection or by direct ELISA against the S protein subunit 1 (S1) of SARS-CoV-2 at the time of blood collection. Blood was collected 4 months post-onset of symptoms (Appendix A). Of the 21 convalescent participants, only one participant had critical illness requiring intubation and immunomodulatory treatments (Appendix A, convalescent-14), whereas the other 20 convalescent participants had mild–moderate disease features. The median age for each group was comparable (Table 1), with an overall median of 38.1 years (±10.7 years). The convalescent group is composed of half women, half men participants, while the vaccinee groups are composed of more women (60% to 80%) than men. White and Asian are the most represented races across all our groups. The minorities most affected by COVID-19 are represented as follows: 20% of our participants are Black/African American, Native, or other races, and 10% are of Hispanic or Latino ethnicity (Table 1).

### 3.2. Anti-S Antibody Responses in Vaccinated and Convalescent Participants

The S protein plays a key role in infectivity and is the immunogenic protein for all four of the US approved COVID-19 vaccines, inducing antibodies, particularly neutralizing antibodies (Nabs), against SARS-CoV-2 [19]. To characterize the antibody responses to the ancestral (wild-type or WT) S protein, we first performed an ELISA to detect anti-S protein subunit 1 (anti-S1) IgG antibody titers in participant serum (Figure 1B). Serum from Moderna vaccinees had significantly higher titers of anti-S1 binding antibodies compared to J&J (*p* < 0.0001) and Pfizer vaccinees (*p* = 0.0036) as well as convalescent participants (*p* < 0.0001). We additionally found that J&J vaccinees had the lowest titers compared to all other groups.

As antibody avidity has been associated with antibody maturation [20], we next analyzed the avidity of these IgG antibodies to ancestral S (Figure 1C) by picking a representative and comparable subset of individuals in each group. The sample size used for this assay was small and the sensitivity of our analysis was limited. Nevertheless, we found no significant differences across the three vaccine groups, but interestingly, convalescent participants trended lower than all three vaccine groups.

We next quantified the plasma neutralizing antibody titers of all four groups against ancestral (SARS-CoV-2 isolate USA-WA1/2020 [WA1]) live virus (Figure 1D) in our immunofluorescence-based microneutralization assay. We found Moderna vaccines had significantly higher neutralizing titers than convalescent participants (*p* = 0.007) and J&J recipients (*p* < 0.0001). The Pfizer group had significantly higher titers than the J&J group (*p* = 0.0055). An overall trend is seen with this rank order: Moderna > Pfizer > convalescent > J&J.

We examined correlations between neutralizing activity levels with S1-binding-IgG levels by Spearman rank correlation coefficient for each group (Figure 1E–H). Interestingly, only participants who received the J&J and Pfizer vaccines showed, respectively, strong and moderate positive correlations of the neutralizing titers with the S1-binding-IgG levels (J&J, Spearman’s r = 0.6753, *p* = 0.0015; Pfizer, Spearman’s r = 0.5017, *p* = 0.0005).

Taken together, the binding Ab titers, neutralizing activity, and IgG avidity results indicate that Moderna vaccinations, relative to Pfizer and J&J vaccinations or natural infections, produced significantly higher magnitudes of S1-binding IgG. This presents a non-significant trend toward higher magnitudes of neutralizing activity in sera, and IgG, which had similar S1 avidity. Correlations of serum neutralizing titers with IgG binding titers were higher and significant for Pfizer and J&J vaccinations.

### 3.3. Spike-Specific T Cell Responses in Vaccinated and Convalescent Participants

We next studied CD4+ T cell responses against the S protein (Figure 2A–C) using an activation-induced marker (AIM) assay, as previously described [18]. Participants’ cryopreserved peripheral blood mononuclear cells (PBMC) were thawed and stimulated overnight using a WT S peptide pool, then analyzed by spectral flow cytometry (Figure 2 and Appendix A). We found no difference by non-parametric Kruskal–Wallis multiple comparisons in the proportion of S peptide-specific AIM+ CD4+ T cells across convalescent and vaccinated participants (Figure 2A). In addition, there was no significant difference in the proportion of S peptide-specific IFN-γ-expressing (Figure 2B) or TNF-α-expressing (Figure 2C) CD4+ T cells.

Similarly, we saw no significant difference in the proportions of S peptide-specific AIM+ CD8+ T cells (Figure 2D), when SARS-CoV-2 spike-specific CD8+ T cells were measured to identify IFN-γ-expressing CD8+ T cells (Figure 2E) or TNF-α-expressing CD8+ T cells (Figure 2F). These data suggest that the context of initial S exposure (vaccination with these platforms versus SARS-CoV-2 infection) has little bearing on S peptide pool activation of the resulting memory CD4+ and CD8+ T cells or their production of certain cytokines.

### 3.4. Spike- and N-Specific Memory B Cell Responses in Vaccinated and Convalescent Participants

Next, we quantified antigen-specific memory B cells (MBC) after the different COVID-19 vaccines or natural infection. We performed an ELISpot MBC assay against the WT receptor binding domain (RBD) (Figure 3A,E), S1 subunit (Figure 3B,E), S2 subunit (Figure 3C,E), or N protein (Figure 3D,E). We found no significant differences across the three vaccine groups and the convalescent group for MBCs against RBD or the S1 subunit. However, there was a significant difference between memory responses against the S2 subunit with convalescent participants showing a significantly higher number of MBCs as compared to the vaccine Moderna group (*p* = 0.0357) and the J&J vaccine group (*p* < 0.0001). Additionally, the vaccine groups showed a lower response to the S2 subunit compared to the S1 subunit (average % S2-specific response vs. average % S1-specific response: 0.17 vs. 0.42 for J&J, 0.34 vs. 0.77 for Moderna, and 0.40 vs. 0.84 for Pfizer), while convalescent participants showed higher memory responses to S2 than S1. As expected, N-response was undetectable for all vaccinee groups. Taken together these data indicate that the vaccine and convalescent groups have generally similar MBC responses to S1 and RBD, but noteworthy differences were identified with S2. S2 MBCs in convalescent patients were of greater magnitude and immunodominant compared to MBCs against the S1 subunit, while the converse was seen with the vaccine recipients.

### 3.5. Antibody Responses against Variants of Concern (VOC) in Vaccinated and Convalescent Participants

Next, we characterized the antibody response against SARS-CoV-2 variants of concern (VOC), including alpha, beta, delta, and omicron BA.1, across our four groups. We first measured avidity for the alpha S1 (Figure 4A), beta S1 (Figure 4B), and delta S1 (Figure 4C). The vaccinated groups showed higher avidity for the alpha S1 than the convalescent participants. The Moderna and Pfizer groups also showed higher avidity for the delta S1 as compared to the convalescent group, with a significant difference between the convalescent and the Pfizer group (*p* = 0.479). These differences suggest broader drift-variant coverage with the mRNA vaccines relative to variant coverage after natural infection.

We also quantified the serum neutralizing titers of all four groups against the delta variant (Figure 4D) and the omicron BA.1 variant (Figure 4E). We found no differences across the various groups and viruses, with the exception that the Moderna delta neutralizing titers were significantly higher than the J&J delta neutralizing titers (*p* = 0.001). While not significant, Moderna participants trended toward higher neutralizing titers compared to the other vaccines and convalescent patients for all three viruses tested. In addition, across all four participant groups, the neutralization titers against WA-1 were higher than the neutralization titers against delta and omicron BA.1 variants (Figure 4F).

## 4. Discussion

During mid-2022 in some parts of the world, such as North America and Europe, the COVID-19 pandemic response has been transforming toward endemicity. Reinfections and breakthrough infections have become common, but are mostly mild to moderate [21,22,23]. Here we explored immunity at 4 months post-vaccination or infection, when it is known to be waning. We aimed to characterize the differences in induced adaptive immunity between the three leading COVID-19 vaccines in use in the US—Moderna mRNA1273, Pfizer-BioNTech BNT162b2, and J&J Ad26.COV2.S—to offer guidance in the ongoing global vaccination effort, as well as to compare them to immunity after natural infection. All the convalescent participants selected for this study were infected between March and April 2020 when the ancestral virus predominated.

While there are several differences in the responses to the three vaccines, we found that these three vaccines all generate relatively similar immune responses in vaccinees at 4 months post-vaccination. This is a reassuring finding for those global communities still largely unvaccinated that may not have a choice in the vaccine(s) they deploy.

In line with the mildly lower reported efficacy against symptomatic disease of the Ad26.COV2.S vaccine [12], we found that the binding antibody levels generated by the J&J vaccine were lower than those of mRNA1273 and BNT162b2 vaccinees. Similar to previous reports [12,24], we also found that the Moderna mRNA vaccine yielded significantly higher binding antibody levels compared to convalescent patients and those vaccinated with either BNT162b2 or Ad26.COV2.S. Moderna participants trended toward higher neutralizing titers compared to the other vaccines and convalescent patients for all three viruses tested, in line with the fact that Moderna participants had the highest binding antibody. The IgG ELISA results did not quite match our findings with live virus neutralization assays, where we found that all three vaccine groups showed similar neutralizing capabilities across ancestral virus, the delta variant, and the omicron BA.1 variant. This indicates that differences in binding titers are not fully predictive of neutralizing titers. Similarly, we found no significant differences across all groups for either CD4+ or CD8+ T cell responses against the ancestral S protein. Taken together, these data underscore the relative similarity of the three vaccines tested here. However, we acknowledge, that our cellular assays used a small sample size, causing the sensitivity of our analysis to be limited.

Previous studies showed vaccinated individuals have different kinetics of antibody levels compared to convalescent patients, with higher initial levels, but a much faster exponential decrease in people who received mRNA vaccines. Individuals vaccinated with mRNA vaccines have shown a continuous decline of their antibody levels over a period of months 4–6 months post-vaccination [25,26,27,28,29,30,31,32]. Individuals vaccinated with Ad26.COV2.S initially elicit substantially lower antibody responses than mRNA vaccines, but their antibody titers increase over the first few months in some individuals [30,33]. In convalescent individuals, antibodies decline during the first few months post-infection, and stabilize between 4–6 months post-infection, with little evidence of decline thereafter [30,34]. Here we focused on a single time point of 4-months post-infection or post-vaccination time point to make a direct comparison of all three vaccines with natural infection.

To control the COVID-19 pandemic, it is essential that vaccination can elicit neutralizing antibodies with broad activity against emerging variants. Unfortunately, previous infection or vaccination with all three vaccines failed to generate a robust and broad neutralizing response against the VOC tested here, with neutralizing titers against VOC significantly lower (from 3.5 to 27.7 folds) relative to titers against the ancestral virus for all groups. This data supports the push for booster vaccination after original vaccination series or vaccination post-infection. Other recent data shows that booster vaccination after the two-doses mRNA vaccine series or double vaccination subsequent to infection yield strong neutralizing titers against those VOC [35].

The higher avidity against VOC in mRNA vaccinees and the immunodominance of S1- over S2-binding antibodies in vaccinees, but the converse in convalescent patients, highlight interesting immunological questions for future study. Why mRNA vaccines might produce broader antibody responses, as suggested by delta S1 avidity data, is not immediately clear. Recent works have shown that mRNA vaccines induce a robust B cell response that leads to persistent germinal center reactions [36,37,38]. Additionally, the immunodominance of S1 over S2 in all vaccinated groups as compared to convalescent patients could have bearing on cross-protective immune responses against future SARS-related coronaviruses, as the S2 subunit contains much of the conserved fusion machinery. While S2 antibody responses are unlikely to be highly neutralizing, this skew could indicate a bias in T cell responses, which would be more capable of generating protective responses against both S1 and S2 peptides displayed by infected cells, however, this hypothesis will require further study.

The data shared here offer a comprehensive look at three of the major vaccines currently being used globally. The similarities between the Moderna mRNA1273, Pfizer-BioNTech BNT162b2, and J&J Ad26.COV2.S vaccines may help to avoid issues of vaccine hesitancy based on preference of one vaccine over another.

In addition, as these vaccines generate similar immunological responses, patients that are unable to use one vaccine or another due to allergies or reported side effects will not be significantly disadvantaged in opting for an alternative vaccine. We hope that these observations will be of value in the continued effort towards global vaccination, in particular, in populations where COVID-19 vaccination coverage remains low.

## 5. Conclusions

Pfizer/BioNTech BNT162b2, Moderna mRNA-1273, and Janssen Ad26.COV2.S COVID-19 vaccines generate similar humoral and cellular immune responses 4 months post-vaccination. All three vaccines elicit a stronger response than natural infection. However, previous infection or vaccination with all three vaccines failed to generate a robust and broad neutralizing response against delta and omicron SARS-CoV-2 variants. The similarity of immune responses from the three vaccines studied here is an important finding in maximizing global protection as vaccination campaigns continue.

## Figures and Tables

**Figure 1 vaccines-10-02152-f001:**
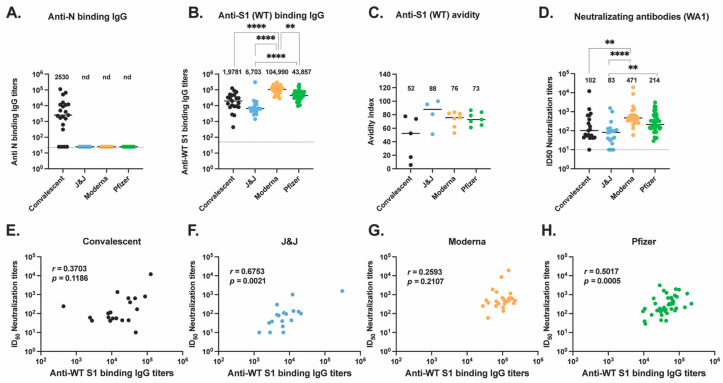
Antibody responses to ancestral SARS-CoV-2 elicited by vaccination or natural infection. (**A**) Anti-N IgG antibody titers (**B**) Anti-S1 IgG antibody titers were assessed for convalescent (black, *n* = 21), J&J (blue, *n* = 20), Moderna (orange, *n* = 28), and Pfizer (green, *n* = 47) participants. (**C**) Anti-S1 (WT) IgG antibody avidity assessed using urea wash ELISA. Data expressed as a ratio of urea-washed absorbance to unwashed absorbance for convalescent (black, *n* = 5), J&J (blue, *n* = 4), Moderna (orange, *n* = 7), and Pfizer (green, *n* = 7) participants. (**D**) Neutralizing antibody titers are shown as log_10_ of half-maximal inhibitory dilution (ID50) for convalescent (black, *n* = 19), J&J (blue, *n* = 18), Moderna (orange, *n* = 25), and Pfizer (green, *n* = 43) participants. For plots A to C, ** *p* < 0.01, and **** *p* < 0.0001 by Kruskal–Wallis test. Dotted line shows limit of detection of the assays. Median is shown by black horizontal bar and values above each group, with data below the limit of detection, marked nd for non-detected. (**E**–**H**) Correlations between neutralizing antibody titers (ID50) and binding antibody titers for the (**E**) convalescent, (**F**) J&J, (**G**) Moderna, and (**H**) Pfizer groups. Spearman’s r and *p* values are shown.

**Figure 2 vaccines-10-02152-f002:**
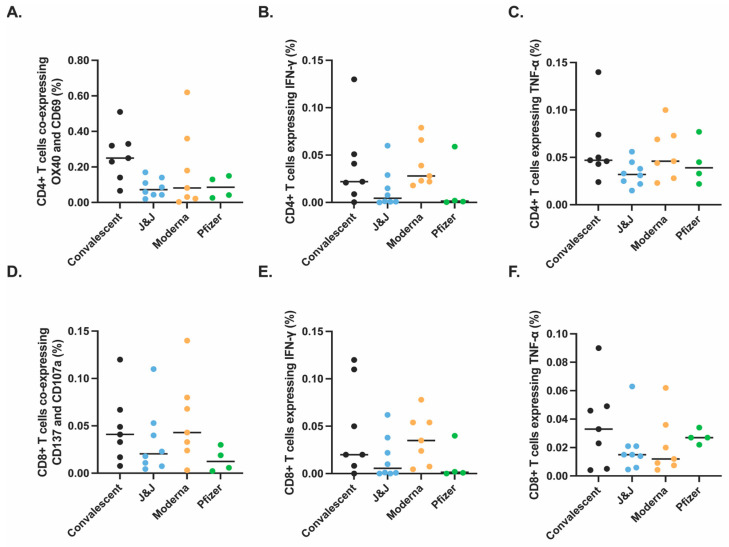
Similar antigen-specific T cells responses observed after vaccination or infection. SARS-CoV-2 specific T cell responses after vaccination or natural infection analyzed by spectral flow cytometry (Appendix A). PBMCs isolated from convalescent (black, *n* = 7), J&J (blue, *n* = 8), Moderna (orange, *n* = 7), and Pfizer (green, *n* = 4) participants were analyzed for SARS-CoV-2 spike-specific CD4+ (**A**–**C**) and CD8+ (**D**–**F**) responses. Significance tested by Kruskal–Wallis test. *y*-axis indicates proportion of total CD4+ or CD8+ T cells that express the indicated activation markers or cytokines.

**Figure 3 vaccines-10-02152-f003:**
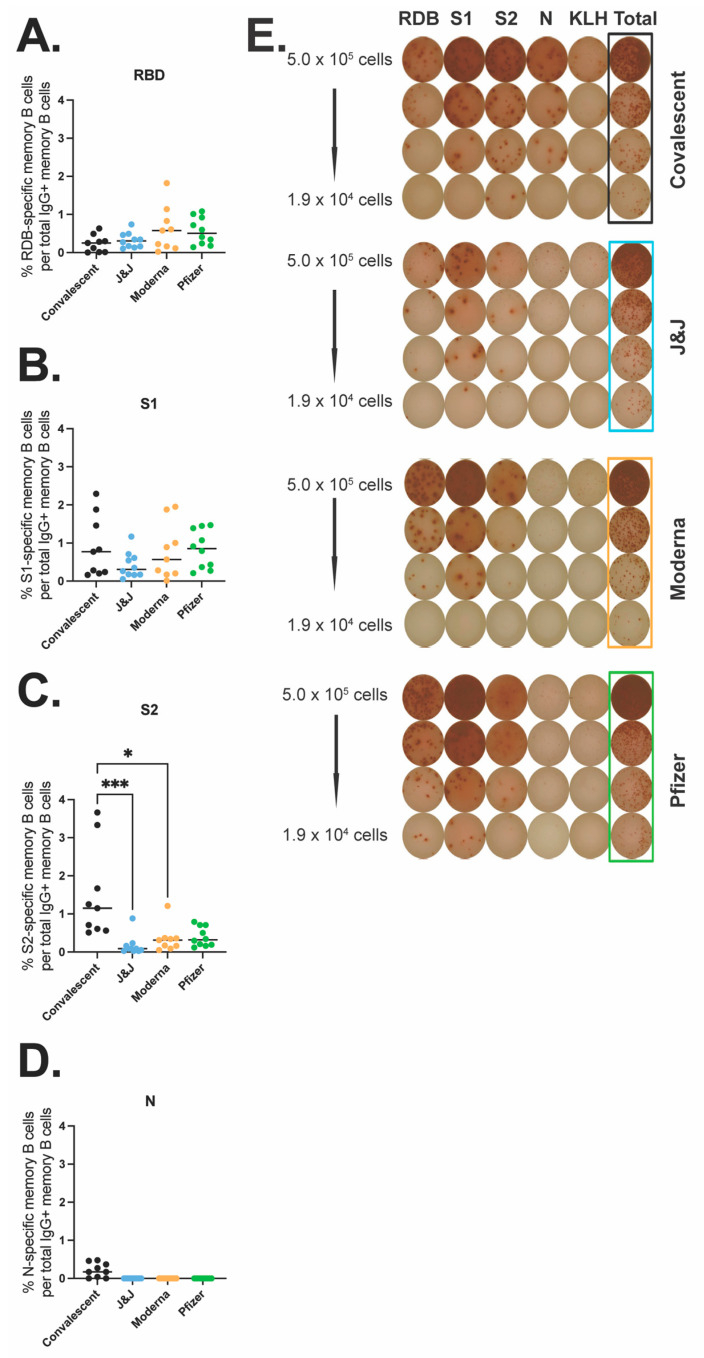
SARS-CoV-2 specific memory B cell responses after vaccination or natural infection. Antigen-specific IgG MBCs from the peripheral blood are shown as a percentage of total IgG-secreting MBCs for RBD (**A**), S1 (**B**), S2 (**C**), and N (**D**) for convalescent (black, *n* = 9), J&J (blue, *n* = 10), Moderna (orange, *n* = 9), and Pfizer (green, *n* = 10) participants. (**E**) Representative MBC ELISpot data, one participant from each group. PBMCs were stimulated with RBD, S1, S2, N protein, or KLH (negative control). * *p* < 0.05 and *** *p* < 0.001 by Kruskal–Wallis.

**Figure 4 vaccines-10-02152-f004:**
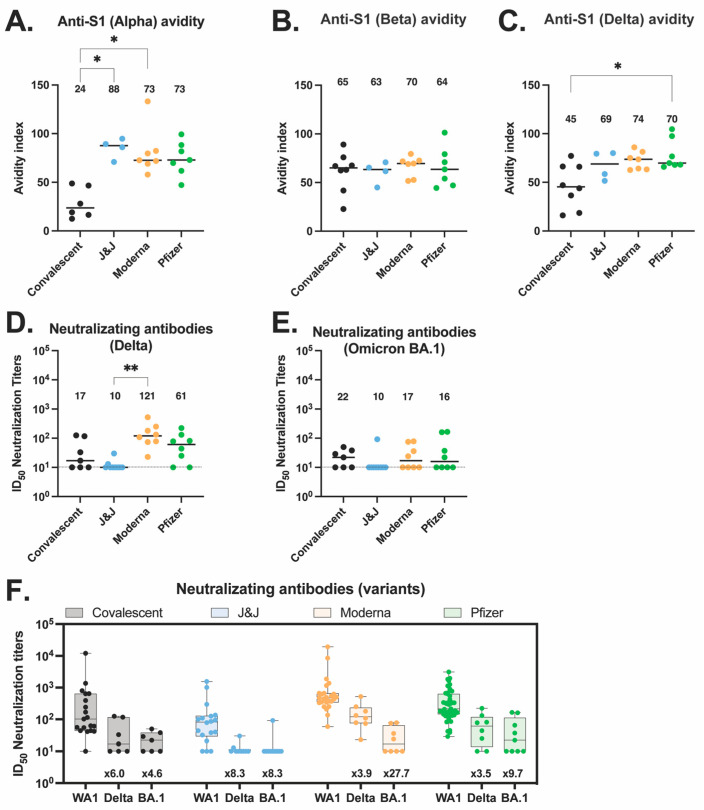
Antibody responses to variants of concerns elicited by vaccination or natural infection. (**A**–**C**) Anti-S1 IgG antibody avidity assessed using urea wash ELISA for alpha (**A**), beta (**B**), and delta (**C**) variants. Data are expressed as a ratio of urea-washed absorbance to unwashed absorbance for convalescent (black, *n* = 6), J&J (blue, *n* = 4), Moderna (orange, *n* = 7), and Pfizer (green, *n* = 7) participants. (**D**) Neutralizing antibody titers against delta variant are shown as log_10_ of half-maximal inhibitory dilution (ID_50_) for convalescent (black, *n* = 7), J&J (blue, *n* = 9), Moderna (orange, *n* = 8), and Pfizer (green, *n* = 8) participants. (**D**,**E**) Neutralizing antibody titers against omicron BA.1 variant are shown as log_10_ of half-maximal inhibitory dilution (ID_50_) for convalescent (black, *n* = 7), J&J (blue, *n* = 10), Moderna (orange, *n* = 8), and Pfizer (green, *n* = 9) participants. For all plots, * *p* < 0.05, and ** *p* < 0.01 by Kruskal–Wallis. Dotted line shows limit of detection of the assays. Median is shown by black horizontal bar and values above each group. (**F**) Summary plot of neutralization titers of delta and omicron BA.1 variants compared to ancestral WA.1 virus for each group. Box plots represent median (horizontal line within the box), and 25th and 75th percentiles (lower and upper border of the box) with individual results depicted with circles. Median are shown above each box plot. The fold change neutralization titers relative to WA-1 are depicted in text at the bottom of the panels.

**Table 1 vaccines-10-02152-t001:** Demographic Characteristics of the Participants in the Study.

Characteristic	Convalescents	J&J	Moderna	Pfizer- BioNTech
	*n* = 21	*n* = 20	*n* = 28	*n* = 47
Sex-no. (%)				
Female	10 (48%)	16 (80%)	22 (79%)	28 (60%)
Male	11 (52%)	4 (20%)	6 (21%)	19 (40%)
Age-yr.	41.1 ± 11.2	36.3 ± 12.5	34.3 ± 9.8	38.7 ± 9.8
Race or ethnic group-no. (%) ^1^				
White	15 (71%)	16 (80%)	19 (68%)	22 (47%)
Asian	3 (14%)	1 (5%)	3 (11%)	20 (43%)
Black or African-American	0 (0%)	1 (5%)	1 (4%)	3 (6%)
Native	0 (0%)	0 (0%)	0 (0%)	0 (0%)
Other	3 (14%)	2 (10%)	5 (18%)	2 (4%)
Hispanic or Latino-no. (%)	2 (10%)	0 (0%)	4 (14%)	5 (11%)
Plus-minus values are median ± SD				

^1^ Race or ethnic group was reported by participants.

## Data Availability

The data presented in this study are available on request from the corresponding author.

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
