# Peer review of "Vaccine-Acquired SARS-CoV-2 Immunity versus Infection-Acquired Immunity: A Comparison of Three COVID-19 Vaccines"

_vaccines, 2022, doi:10.3390/vaccines10122152_

Round 1

Reviewer 1 Report

Samanovic et al analyzed compared post-immunization immune responses from 95 individuals who had completed either the Moderna, Pfeizer, or J&J SARS-CoV-2 immunization(s) and 21 Covid19 convalescent individuals. The paper is well written, interesting to read, and relevant to the scientific community. I just have some minor comments.

1) line 237: remove first comma in the sentence

2) Fig 1C: for this figure, the authors selected "a representative and comparable subset of individuals in each group," but this resulted in very few data points for each group (7 or less). A way to better validate the trends shown in this plot would be to repeat the tests multiple times, each time with different groups chosen at random and then assess via a bootstrap analysis. Is the data sufficiently diverse for this kind of sensitivity analysis? If yes, I recommend adding, if not I recommend the authors state why and add this caveat to the main text. 

3) line 276: Participants is missing an apostrophe

4) Figure 2 has also the low-sample size potential issues mentioned earlier .. is this because of low availability of participants' PBMC? 

Finally, it would be interesting to know when during the pandemic the convalescent participants had been infected, and also where ... mostly because I wonder if the authors, based on this information, are able to infer that they were all infected with the wild type Wuhan variant or if they could have been exposed to a later variant. I think this would be relevant to add to the interpretation of the data presented in section 3.5 in particular.

Author Response

Thank you for your review of our manuscript and your comments. 

1) line 237: remove first comma in the sentence. Comma has been removed.

2) Fig 1C: for this figure, the authors selected "a representative and comparable subset of individuals in each group," but this resulted in very few data points for each group (7 or less). A way to better validate the trends shown in this plot would be to repeat the tests multiple times, each time with different groups chosen at random and then assess via a bootstrap analysis. Is the data sufficiently diverse for this kind of sensitivity analysis? If yes, I recommend adding, if not I recommend the authors state why and add this caveat to the main text. The avidity assay was used here as a preliminary analysis of the overall strength of the interactions between the serum antibodies and the variant antigens. We agree that only a small subset of participants was tested for this analysis and this is a limitation of our analysis. However, we completed that first study with neutralization assays with live virus on a larger set of participants. The neutralization assay confirmed the data obtained in the preliminary analysis of the avidity assay. The following comment was included to the paragraph line 246:

As antibody avidity has been associated with antibody maturation [20], we next analyzed the avidity of these IgG antibodies to ancestral S (Figure. 1C) by picking a representative and comparable subset of individuals in each group. The sample size used for this assay was small and the sensitivity of our analysis was limited. Nevertheless, we found no significant differences across the three vaccine groups, but interestingly convalescent participants trended lower than all three vaccine groups.

3) line 276: Participants is missing an apostrophe. Apostrophe was added.

4) Figure 2 has also the low-sample size potential issues mentioned earlier .. is this because of low availability of participants' PBMC?  Due to the limitation in available PBMC, only a subset of participants for each group was analyzed. The AIM assay requires over 10 million PBMCs for each assay. The subset in each group was chosen depending on PBMC availability and comparable characteristics (demographic and clinical history).

Finally, it would be interesting to know when during the pandemic the convalescent participants had been infected, and also where ... mostly because I wonder if the authors, based on this information, are able to infer that they were all infected with the wild type Wuhan variant or if they could have been exposed to a later variant. I think this would be relevant to add to the interpretation of the data presented in section 3.5 in particular. All the convalescent participants selected for this analysis, were infected between March and April 2020 when ancestral virus predominated. All blood samples were collected in NYC, however, participants may have been infected during travel time. At this early time of the pandemic, only few mutations had appeared and been identified worldwide in SARS-CoV-2. New variants of concerns were only identified at the end of 2020. Thus, we can confidently hypothesize that those individuals were all infected with a SARS-CoV-2 virus closely related to the ancestral Wuhan lineage. The following statement was added on line 405 of the discussion:

“All the convalescent participants selected for this study were infected between March and April 2020 when ancestral virus predominated.”

See attachement.

Reviewer 2 Report

1.It is suggested to add new in vivo research studies about the toxicological concerns.
2.what is the suggestion of this study for future works?
3.Please discuss and compare your results with previous works
4.It will be better to add the role of mitochondria.
5.Please add details for time period and dose selection.
5.More references for the discussion part of manuscript and bold your study novelty should be added: e.g.,
-DOI: 10.1016/j.micinf.2022.105077
-DOI: 10.2217/nnm-2020-0441

Author Response

Thank you for your review of our manuscript and your comments.

1.It is suggested to add new in vivo research studies about the toxicological concerns.

We direct the reviewer to the results from the clinical trials for the various US vaccines which give a full overview of possible side effects in humans. We respectfully consider that additional safety studies are beyond the scope of this study and unnecessary.

2.what is the suggestion of this study for future works?

The higher avidity against VOC in mRNA vaccinees and the immunodominance of S1- over S2-binding antibodies in vaccinees, but the converse in convalescent patients, highlight interesting immunological questions for future study. Why mRNA vaccines might produce broader antibody responses, as suggested by delta S1 avidity data, is not immediately clear. Recent works have shown that mRNA vaccines induce a robust B cell response that leads to persistent germinal center reactions [36-38].

Additionally, the immunodominance of S1 over S2 in all vaccinated groups as compared to convalescent patients could have bearing on cross-protective immune responses against future SARS-related coronaviruses, as the S2 subunit contains much of the conserved fusion machinery. While S2 antibody responses are unlikely to be highly neutralizing, this skew could indicate a bias in T cell responses, which would be more capable of generating protective responses against both S1 and S2 peptides displayed by infected cells however this hypothesis will require further study.

3. Please discuss and compare your results with previous works

In the discussion the following points are discussed and added:

"In line with the mildly lower reported efficacy against symptomatic disease of the Ad26.COV2.S vaccine [12], we found that the binding antibody levels generated by the J&J vaccine were lower than those of mRNA1273 and BNT162b2 vaccinees. Similar to previous reports [12, 24], we also found that the Moderna mRNA vaccine yielded significantly higher binding antibody levels compared to convalescent patients and those vaccinated with either BNT162b2 or Ad26.COV2.S."

"Previous studies showed vaccinated individuals have different kinetics of antibody levels compared to convalescent patients, with higher initial levels, but a much faster exponential decrease in people who received mRNA vaccines. Individuals vaccinated with mRNA vaccines have shown a continuous decline of their antibody levels over a period of months 4-6 months post-vaccination [25-32]. Individuals vaccinated with Ad26.COV2.S initially elicit substantially lower antibody responses than mRNA vaccines, but their antibody titers increase over the first few months in some individuals [30, 33]. In convalescent individuals, antibodies decline during the first few months post-infection, and stabilize between 4-6 months post-infection, with little evidence of decline thereafter [30, 34]. Here we focused on a single time point 4-months post-infection or post-vaccination time point to make a direct comparison of all three vaccines with natural infection."

4.It will be better to add the role of mitochondria.

Such cell biology studies are beyond the scope of this study to evaluate the general immune responses to approved COVID vaccines.

5.Please add details for time period and dose selection.

The following is included in the materials and methods, line 79, to detail time of collection.

Participants had blood drawn around four months post-vaccination (post-second dose for mRNA vaccines, post-1st dose for Ad26.COV2.S vaccine) or post-COVID-19 (post-onset of symptoms). The median numbers of days ± standard deviations were: 124 ± 12 for BNT162b2; 126 ± 15 for mRNA-1273, 118 ± 20 for Ad26.COV2.S, and 120 ± 20 for convalescent participants.

5.More references for the discussion part of manuscript and bold your study novelty should be added: e.g.,
-DOI: 10.1016/j.micinf.2022.105077
-DOI: 10.2217/nnm-2020-0441

The following suggested citation was added to the discussion:

Jacot. Microbes Infect. 2022 Nov 15;105077. doi: 10.1016/j.micinf.2022.105077.

Along with the following statement regarding our Nab results with VOC, line 445: “This data supports the push for booster vaccination after original vaccination series or vaccination post-infection. Other recent data shows that booster vaccination after the two-doses mRNA vaccine series or double vaccination subsequent to infection yield strong neutralizing titers against those VOC [35].”

See attachement.

Reviewer 3 Report

Great paper, honestly. I couldn't find much wrong other than the couple tiny things below, but they're inconsequential. Top tier work and an excellent presentation of your data.

47: provides

237: extra comma after “As”

Figure 2: figure legend doesn’t mention flow cytometry. Not that you have to, but figure 1 mentions ELISA and 3 mentions ELISpot

Author Response

Thank you for your review of our manuscript and your comments.

47: provides. This was corrected in the text.

237: extra comma after “As” Comma has been removed.

Figure 2: figure legend doesn’t mention flow cytometry. Not that you have to, but figure 1 mentions ELISA and 3 mentions ELISpot. For Figure 2, “analyzed by spectral flow cytometry” was added to line 296.

Reviewer 4 Report

The authors did wet lab experiments to compare the immune responses generated by the three most widely used COVID-19 vaccines in the USA and in infected individuals. Strengths of the study include i) that four scenarios were compared ii) that the authors evaluated both B cell and T cell responses iii) evaluation of responses to some COVID-19 variants and iv) that the manuscript is very well written, except for the paucity of references. Weaknesses of the study include v) small sample size and that vi) immune responses were measured only at one snapshot approximately four months after infection/vaccination. The main implication of the study is that in areas of the   world where vaccination has been slow, any of the Pfizer, Moderna, and Johnson & Johnson vaccines would be useful, but the Johnson & Johnson vaccine is inferior to the two mRNA vaccines.

Major concerns:

1.              Various statements on page 6 and elsewhere assess whether there is or is not a significant difference in two distributions of immune responses (e.g., the responses between recipients of two different vaccines). The legends of Figures 2 and 3 indicate that the statistical comparisons were done by one-way ANOVA, which is a reasonable choice if what matters is the mean response. However, some of the distributions of immune response plotted in Figures 2 and 3 are quite broad (i.e., large range of values on the y-axis) and include dots with a quite low y-axis value. Moreover, for clinical purposes what probably matters is what proportion of individuals have an immune response above some threshold rather than the mean response. As such, the study is grossly underpowered to characterize the lower tails of the immune response distributions. The authors should i) explain why ANOVA is or is not the best test to use, ii) re-evaluate the results in Figures 2 and 3 using an alternative statistical test that focuses on the lower tails and iii) acknowledge in the Discussion the limitation that the small sample size makes it difficult to assess what proportion of vaccinees mount an insufficient immune response either in B cells or in T cells.

2.              Line 60-61: “All three vaccines elicit a stronger 60 response than natural infection.” needs to be clarified at length because of the time course. Various studies (see suggested references below) have shown that the humoral response in vaccinated individuals declines much more rapidly than in infected individuals. So, the comparison at 4 months is just a snapshot and not sufficiently indicative of the immune response after 6 or more months.

3.              All occurrences of the words “significant” or “significantly” should be accompanied by a p-value in parentheses.

Minor concerns:

4.              Line 38, need to define “fully vaccinated individuals” because the meaning of the quoted term differs between the mRNA vaccines (two doses) and the Johnson & Johnson vaccine (one does)

5.              The passage at lines 227-231 that starts “SARS-CoV-2 has four key structural proteins:… should be moved to the Introduction because multiple sentences occurring before line 227 refer to the S-protein or to the N-protein. In addition, the authors should add text to define what they mean by “S1 subunit” and “S2 subunit”.

6.              The sentence “Reinfections and breakthrough infections have become common, but are mostly mild to moderate.” at lines 358-359 should be justified by multiple references.

I suggest to add (citations to) the following references to address comments 2 and 6 above. The authors are encouraged to find and cite additional references regarding either the time course of immune responses (comment 2) or the natural history of reinfections (comment 6).

Suggested References to Add:

L. A. dos Santos et al. Recurrent COVID-19 including evidence of reinfection and enhanced severity in thirty Brazilian healthcare workers. Journal of Infection 2021; 82:399-406.

A.D. Inchingolo et al. Antispike immunoglobulin-G (IgG) titer response of SARS-CoV-2 mRNA-vaccine (BNT162b2): A monitoring study on healthcare workers. Biomedicines 2022; 10:2402.

A.     Israel et al. Large-scale study of antibody titer decay following BNT162b2 mRNA vaccine or SARS-CoV-2 infection. Vaccines 2022; 10:64.

T. Lagousi et al. Comparative characterization of human antibody response induced by BNT162b2 vaccination vs. SARS-CoV-2 wild-type infection. Vaccines 2022; 19:1210.

X. Ren et al. Reinfection in patients with COVID-19: a systematic review. Global Health Research and Policy 2022; 7:12.

A.    T. Roberts et al. Reinfection of SARS-CoV-2 – analysis of 23 cases from the literature. Infectious Diseases 2021; 7:479-485.

A.  Sette, S. Crotty. Immunological memory to SARS-CoV-2 infection and COVID-19 vaccines. Immunological Reviews 2022; 310:27-46.

C. J. Toro-Huamanchumo et al. Clinical and epidemiological features of patients with COVID-19 reinfection: a systematic review. New Microbes and New Infections 2022; 48:101021.

N. Zacks et al. Assessment of Predictors for SARS-CoV-2 Antibodies Decline Rate in Health Care Workers after BNT162b2 Vaccination - Results from a Serological Survey. Vaccines 2022; 10: 1443.

Author Response

Thank you for your review of our manuscript and your comments.

1. i) We agree with the reviewer comments. An assumption/requirement of a one-way ANOVA is that the data are normally distributed. We reanalyzed our data using a non-parametric ANOVA (Kruskal-Wallis), and modified the text accordingly. The overall results and interpretation of our data remain the same.

ii) For Figure 3, we performed a subanalysis with thresholding, as suggested, using IFNg- and TNF-markers  as thresholding standards. Our thresholds values were calculated from data previously obtained with the same AIM protocol performed on PBMC of SARS-CoV-2 naïve individuals at baseline pre-vaccination (published data in Samanovic MI, et al. Sci Transl Med. PMID: 34874183). After exclusion of the values below threshold, statistical analysis gave similar results (non-significance difference between groups).

iii) The limitation of analysis of the T and B cell responses is now acknowledged in the discussion.

2. The following sentence was added, to clarify our conclusion, line 67:

“All three vaccines elicit a stronger response than natural infection 4 months post-vaccination or exposure to the virus.”

Due to constraints on sample availability, we did not analyze later time points for longitudinal responses. We have made clear throughout the manuscript that this is focused on the single time point; subsequent studies will be needed to analyze antibody kinetics and comparisons of levels at alternative time points.

3. P-values were added for all occurrences of the words “significant” or “significantly.

4. The following was added to line 44 to defined “fully vaccinated individuals” 

“(defined as 2 weeks after one dose of a J&J single-dose vaccine, or 2 weeks after a second dose of an mRNA 2-dose vaccine series)”

5. The mentioned passage on line 227, was moved to the introduction. In addition, the following text about S1 and S2 was added for clarity:

“The S protein plays a key role in the receptor recognition and cell membrane fusion process and is the immunogenic protein for all four of the US approved COVID-19 vaccines. The S protein is composed of two subunits: S1 (containing a receptor-binding domain) and S2 (mediating viral cell membrane fusion).”

6. The following references were added to justify the sentence:

A. dos Santos et al. Recurrent COVID-19 including evidence of reinfection and enhanced severity in thirty Brazilian healthcare workers. Journal of Infection 2021; 82:399-406.

J. Toro-Huamanchumo et al. Clinical and epidemiological features of patients with COVID-19 reinfection: a systematic review. New Microbes and New Infections 2022; 48:101021.

Ren et al. Reinfection in patients with COVID-19: a systematic review. Global Health Research and Policy 2022; 7:12.

See attachement.
